# ARE NEURAL RANKERS STILL OUTPERFORMED BY GRADIENT BOOSTED DECISION TREES?

**Zhen Qin, Le Yan, Honglei Zhuang, Yi Tay, Rama Kumar Pasumarthi,**
**Xuanhui Wang, Michael Bendersky, Marc Najork**
Google Research
{zhenqin,lyyanle,hlz,yitay,ramakumar,xuanhui,bemike,najork}@google.com

## ABSTRACT

Despite the success of neural models on many major machine learning problems, their effectiveness on traditional Learning-to-Rank (LTR) problems is still not widely acknowledged. We first validate this concern by showing that most recent neural LTR models are, by a large margin, inferior to the best publicly available Gradient Boosted Decision Trees (GBDT) in terms of their reported ranking accuracy on benchmark datasets. This unfortunately was somehow overlooked in recent neural LTR papers. We then investigate why existing neural LTR models under-perform and identify several of their weaknesses. Furthermore, we propose a unified framework comprising of counter strategies to ameliorate the existing weaknesses of neural models. Our models are the first to be able to perform equally well, comparing with the best tree-based baseline, while outperforming recently published neural LTR models by a large margin. Our results can also serve as a benchmark to facilitate future improvement of neural LTR models.

## 1 INTRODUCTION

Neural approaches have been dominating in many major machine learning domains, such as computer vision (He et al., 2015), natural language processing (Devlin et al., 2019), and speech recognition (Hannun et al., 2014). However, the effectiveness of neural approaches in traditional Learning-to-Rank (LTR), the long-established inter-disciplinary research area at the intersection of machine learning and information retrieval (Liu, 2009), is not widely acknowledged (Yang et al., 2019), especially on benchmark datasets that have only numerical features.

Historically, a series of LTR models were developed by researchers at Microsoft, starting with RankNet (Burges et al., 2005) and LambdaRank (Burges et al., 2007), both based on neural networks, and culminating in LambdaMART (Wu et al., 2010), which is based on Gradient Boosted Decision Trees (GBDT); Burges (2010) provides an overview of this evolution. There are two publicly available implementations of LambdaMART: one provided by the RankLib[1] library that is part of the Lemur Project (henceforth referred to as $\lambda\text{MART}_{RankLib}$); and the LightGBM[2] implementation provided by Microsoft (Ke et al., 2017) (henceforth referred to as $\lambda\text{MART}_{GBM}$). As we will show in Section 3, $\lambda\text{MART}_{GBM}$ substantially outperforms $\lambda\text{MART}_{RankLib}$.

There is strong and continuing interest in neural ranking models, with numerous papers published in the last few years alone. Most of these papers treat RankNet and LambdaRank as weak baselines (Pang et al., 2020; Bruch et al., 2019b) and LambdaMART as the "state-of-the-art" (Bruch et al., 2019b; Li et al., 2019; Zhu & Klabjan, 2020; Hu et al., 2019). However, when examining these papers, we note that they either acknowledge their under-performance to $\lambda\text{MART}_{GBM}$ or claim state-of-the-art performance by comparing to a weaker $\lambda\text{MART}_{RankLib}$ implementation. The inconsistency of performance evaluation on benchmark datasets in this field has made it difficult to measure progress (Lipton & Steinhardt, 2018). It therefore remains an open question whether neural LTR models are as effective as they claim to be, and how to improve them if that is not the case.

---

[1] https://sourceforge.net/p/lemur/wiki/RankLib/
[2] https://github.com/microsoft/LightGBM

In this paper, we first conduct a benchmark to show that $\lambda\text{MART}_{GBM}$ outperforms recently published neural models, as well as the $\lambda\text{MART}_{RankLib}$, by a **large** margin. While the neural paradigm is still appealing in a myriad of ways, such as being composable, flexible, and able to benefit from a plethora of new advances (Vaswani et al., 2017; Devlin et al., 2019), the research progress in neural ranking models could be hindered due to their inferior performance to tree models. It thus becomes critical to understand the pitfalls of building neural rankers and boost their performance on benchmark datasets.

Specifically, we investigate why neural LTR approaches under-perform on standard LTR datasets and identify three major weaknesses that are typically ignored by recent work. First, neural models are not as adept at performing effective feature transformations and scaling, which is one major benefit of using tree-based methods (Saberian et al., 2019). In ranking data which is typically long-tailed, this can be a prohibitive property. Second, standard feed-forward networks are ineffective in generating higher-order features as noted by recent papers (Wang et al., 2017b; Beutel et al., 2018). More effective network architectures for neural LTR models are needed. Third, recent neural LTR work on benchmark datasets does not employ high-capacity networks, a key success factor of many neural models (Devlin et al., 2019), possibly due to a small scale of training data that causes overfitting. On the other hand, there are several potential benefits of neural approaches over LambdaMART for LTR, such as their flexibility to model listwise data and the existence of many techniques to mitigate data sparsity. To that end, we propose a new framework that ameliorates the weaknesses of existing neural LTR approaches and improves almost all major network components.

In the proposed framework, we make several technical contributions: (1) We demonstrate empirical evidence that a simple log1p transformation on the input features is very helpful. (2) We use data augmentation (DA) to make the most out of high-capacity neural models, which is surprisingly the first work in the LTR literature to do so. We show that adding a simple Gaussian noise helps, but only when the model capacity is appropriately augmented (which probably explains why there is no prior work on such a simple idea). (3) We use self-attention (SA) to model the listwise ranking data as context, and propose to use latent cross (LC) to effectively generate the interaction of each item and its listwise context.

We conduct experiments on three widely used public LTR datasets. Our neural models are trained with listwise ranking losses. On all datasets, our framework can outperform recent neural LTR methods by a large margin. When comparing with the strong LambdaMART implementation, $\lambda\text{MART}_{GBM}$, we are able to achieve equally good results, if not better. Our work can also serve as a benchmark for neural ranking models, which we believe can lay a fertile ground for future neural LTR research, as rigorous benchmarks on datasets such as ImageNet (Russakovsky et al., 2015) and GLUE (Wang et al., 2018a) do in their respective fields.

## 2 BACKGROUND

We provide some background on LTR, including its formulation and common metrics. We review LambdaMART and highlight its two popular implementations which are causes of the inconsistency of evaluations in the recent literature.

### 2.1 LEARNING TO RANK

LTR methods are supervised techniques and the training data can be represented as a set $\Psi = \{(\mathbf{x}, \mathbf{y}) \in \chi^n \times \mathbb{R}^n)\}$, where $\mathbf{x}$ is a list of $n$ items $x_i \in \chi$ and $\mathbf{y}$ is a list of $n$ relevance labels $y_i \in \mathbb{R}$ for $1 \leq i \leq n$. We use $\chi$ as the universe of all items. In traditional LTR problems, each $x_i$ corresponds to a query-item pair and is represented as a feature vector in $\mathbb{R}^k$ where $k$ is the number of feature dimensions. With slightly abuse of notation, we also use $x_i$ as the feature vector and say $\mathbf{x} \in \mathbb{R}^{n \times k}$. The objective is to learn a function that produces an ordering of items in $\mathbf{x}$ so that the utility of the ordered list is maximized.

Most LTR algorithms formulate the problem as learning a ranking function to score and sort the items in a list. As such, the goal of LTR boils down to finding a parameterized ranking function

$s(\cdot; \Theta) : \chi^n \to \mathbb{R}^n$, where $\Theta$ denotes the set of parameters, to minimize the empirical loss:

$$\mathcal{L}(s) = \frac{1}{|\Psi|} \sum_{(\mathbf{x}, \mathbf{y}) \in \Psi} l(\mathbf{y}, s(\mathbf{x})), \tag{1}$$

where $l(\cdot)$ is the loss function on a single list. LTR algorithms differ primarily in how they parameterize $s$ and how they define $l$.

There are many existing ranking metrics such as NDCG and MAP used in LTR problems. A common property of these metrics is that they are rank-dependent and place more emphasis on the top ranked items. For example, the commonly adopted NDCG metric is defined as

$$NDCG(\pi_s, \mathbf{y}) = \frac{DCG(\pi_s, \mathbf{y})}{DCG(\pi^*, \mathbf{y})}, \tag{2}$$

where $\pi_s$ is a ranked list induced by the ranking function $s$ on $\mathbf{x}$, $\pi^*$ is the ideal list (where $\mathbf{x}$ is sorted by $\mathbf{y}$), and $DCG$ is defined as:

$$DCG(\pi, \mathbf{y}) = \sum_{i=1}^{n} \frac{2^{y_i} - 1}{\log_2(1 + \pi(i))} = \sum_{i=1}^{n} \frac{G_i}{D_i} \tag{3}$$

In practice, the truncated version that only considers the top-k ranked items, denoted as NDCG@k, is often used.

## 2.2 LAMBDAMART

LTR models have evolved from linear models (Joachims, 2002), to nerual networks (Burges et al., 2005), and then to decision trees (Burges, 2010) in the past two decades. LambdaMART, proposed about ten years ago (Wu et al., 2010; Burges, 2010), is still treated as the "state-of-the-art" for LTR problems in recent papers (Bruch et al., 2019b; Zhu & Klabjan, 2020). It is based on Gradient Boosted Decision Trees (GBDT). During each boosting step, the loss is dynamically adjusted based on the ranking metric in consideration. For example, $\Delta$NDCG is defined as the absolute difference between the NDCG values when two documents $i$ and $j$ swap their positions in the ranked list sorted by the obtained ranking functions so far.

$$\Delta NDCG(i, j) = |G_i - G_j| \cdot \left| \frac{1}{D_i} - \frac{1}{D_j} \right|. \tag{4}$$

Then LambdaMART uses a pairwise logistic loss and adapts the loss by re-weighting each item pair in each iteration, with $s(\mathbf{x})|_i$ being the score for item $i$ and $\alpha$ being a hyperparameter:

$$l(\mathbf{y}, s(\mathbf{x})) = \sum_{y_i > y_j} \Delta NDCG(i, j) \log_2(1 + e^{-\alpha(s(\mathbf{x})|_i - s(\mathbf{x})|_j)}) \tag{5}$$

There are two popular public implementations of LambdaMART, namely $\lambda$MART$_{GBM}$ and $\lambda$MART$_{RankLib}$. $\lambda$MART$_{GBM}$ is more recent than $\lambda$MART$_{RankLib}$ and has more advanced features by leveraging novel data sampling and feature bundling techniques (Ke et al., 2017). However, recent neural LTR papers either use the weaker implementation of $\lambda$MART$_{RankLib}$ (Pang et al., 2020; Wang et al., 2017a; Ai et al., 2018; 2019), or acknowledge the inferior performance of neural models when compared with $\lambda$MART$_{GBM}$ (Bruch et al., 2019b). Such an inconsistency makes it hard to determine whether neural models are indeed more effective than the tree-based models.

## 3 BENCHMARKING EXISTING METHODS

To resolve the inconsistency, we perform a benchmark on three popular LTR benchmark datasets to show that: 1) there is a large gap between the two implementations of tree-based LambdaMART $\lambda$MART$_{GBM}$ and $\lambda$MART$_{RankLib}$; 2) Recent neural LTR methods are generally significantly worse than the stronger implementation. Then we discuss several weaknesses of recent neural LTR approaches, and point out promising directions, which lay the foundation of our proposed framework.

Table 1: The statistics of the three largest public benchmark datasets for LTR models.

|  | #features | #queries | | | #docs | | |
|---|---|---|---|---|---|---|---|
|  |  | training | validation | test | training | validation | test |
| Web30K | 136 | 18,919 | 6,306 | 6,306 | 2,270,296 | 747,218 | 753,611 |
| Yahoo | 700 | 19,944 | 2,994 | 6,983 | 473,134 | 71,083 | 165,660 |
| Istella | 220 | 20,901 | 2,318 | 9,799 | 6,587,822 | 737,803 | 3,129,004 |

Table 2: All numbers are significantly worse than the corresponding number from $\lambda\text{MART}_{GBM}$ at the $p < 0.05$ level using a two-tailed $t$-test. Best performing numbers are bold.

| Models | Rerank | Web30K NDCG@k | | | Yahoo NDCG@k | | | Istella NDCG@k | | |
|---|---|---|---|---|---|---|---|---|---|---|
|  |  | @1 | @5 | @10 | @1 | @5 | @10 | @1 | @5 | @10 |
| $\lambda\text{MART}_{RankLib}$ | ✗ | 45.35 | 44.59 | 46.46 | 68.52 | 70.27 | 74.58 | 65.71 | 61.18 | 65.91 |
| $\lambda\text{MART}_{GBM}$ | ✗ | **50.73** | **49.66** | **51.48** | **71.88** | **74.21** | **78.02** | **74.92** | **71.24** | **76.07** |
| RankSVM | ✗ | 30.10 | 33.50 | 36.50 | 63.70 | 67.40 | 72.60 | 52.69 | 50.41 | 55.29 |
| GSF | ✗ | 41.29 | 41.51 | 43.74 | 64.29 | 68.38 | 73.16 | 62.24 | 59.68 | 65.08 |
| ApproxNDCG | ✗ | 46.64 | 45.38 | 47.31 | 69.63 | 72.32 | 76.77 | 65.81 | 62.32 | 67.09 |
| DLCM | ✓ | 46.30 | 45.00 | 46.90 | 67.70 | 69.90 | 74.30 | 65.58 | 61.94 | 66.80 |
| SetRank | ✗ | 42.90 | 42.20 | 44.28 | 67.11 | 69.60 | 73.98 | 67.33 | 62.78 | 67.37 |
| $\text{SetRank}^{re}$ | ✓ | 45.91 | 45.15 | 46.96 | 68.22 | 70.29 | 74.53 | 67.60 | 63.45 | 68.34 |

## 3.1 DATASETS

The three data sets we used in our experiments are public benchmark datasets widely adopted by the research community. They are the LETOR dataset from Microsoft (Qin & Liu, 2013), Set1 from the YAHOO LTR challenge (Chapelle & Chang, 2011), and Istella (Dato et al., 2016). We call them Web30K, Yahoo, and Istella respectively. All of them are data sets for web search ranking and the largest data sets publicly available for LTR algorithms. The relevance labels of documents for each query are rated by human in the form of multilevel graded relevance. See Qin & Liu (2013) for an example list of features, such as the number of URL clicks, or the BM25 scores of the different page sections. An overview of these three datasets is shown in Table 1.

## 3.2 COMPARISON

We compare a comprehensive list of methods in Table 2. $\lambda\text{MART}_{GBM}$ (Ke et al., 2017) and $\lambda\text{MART}_{RankLib}$ are the two LambdaMART implementations. RankSVM (Joachims, 2006) is a classic pairwise learning-to-rank model built on SVM. GSF (Ai et al., 2019) is a neural model using groupwise scoring function and fully connected layers. ApproxNDCG (Bruch et al., 2019b) is a neural model with fully connected layers and a differeiable loss that approximates NDCG (Qin et al., 2010). DLCM (Ai et al., 2018) is an RNN based neural model that use list context information to rerank a list of documents based on $\lambda\text{MART}_{RankLib}$ as in the original paper. SetRank (Pang et al., 2020) is a neural model using self-attention to encode the entire list and perform a joint scoring. $\text{SetRank}^{re}$ (Pang et al., 2020) is SetRank plus ordinal embeddings based on the initial document ranking generated by $\lambda\text{MART}_{RankLib}$ as in the original paper.

We choose to compare these methods because they are either popular or recent. The neural models are already leveraging advanced neural techniques such as using neural methods to model the entire ranking list, which is difficult for tree-based models to achieve. We reproduced results for $\lambda\text{MART}_{RankLib}$, $\lambda\text{MART}_{GBM}$, RankSVM, GSF, and ApproxNDCG with extensive hyperparameter tuning with more details in Appendix A. Results for the DLCM and SetRank methods are from their respective papers where the authors did their own tuning. Note that the test set is fixed for all datasets, thus the numbers are comparable.

From Table 2, we can see the following. 1) $\lambda\text{MART}_{GBM}$ is a more appropriate "state-of-the-art" LambdaMART baseline, as it significantly outperforms $\lambda\text{MART}_{RankLib}$. 2) Recent neural LTR methods, though sometimes outperform $\lambda\text{MART}_{RankLib}$, are inferior to $\lambda\text{MART}_{GBM}$ by a large margin,

sometimes by as much as 15%, comparatively. These results show the inconsistency of existing methods and validate the concerns on the current practice of neural LTR models[3].

## 4 NEURAL LTR MODELS

A natural question is: why do neural models under-perform on LTR benchmark datasets compared with LambdaMART, despite their success in many machine learning research areas? We first identify a few weaknesses of the neural LTR models and then propose our methods to address them.

### 4.1 WEAKNESSES

By reviewing recent papers and the strength of tree-based models, we give the following hypotheses:

**Feature transformation**. Neural networks are sensitive to input feature scales and transformations (Saberian et al., 2019). LTR datasets consist of features of diverse scales with long-tail distributions, such as the number of clicks of an item. Tree-based models are known to partition the feature space effectively, which is beneficial for datasets (such as LTR datasets) with only numeric features. Some recent work already shows the benefits of better input feature transformations than Gaussian normalization (Saberian et al., 2019; Zhuang et al., 2020). Unfortunately, neither the pioneering neural LTR papers (Burges et al., 2005; 2007) nor the most recent ones discuss the impact of feature transformation.

**Network architecture**. Unless the focus is the neural architecture, neural LTR papers typically use a standard feed-forward network that consists of a stack of fully connected layers. However, fully connected layers are known to be ineffective in generating higher-order feature interactions. The problem has been widely studied in areas such as ads prediction (Wang et al., 2017b) and recommender systems (Beutel et al., 2018), but has not received enough attention for LTR.

**Data sparsity**. Recent neural LTR models are small and do not employ high-capacity networks (Bruch et al., 2019b; Pang et al., 2020), possibly due to the overfitting issue. While large datasets are key factors to many recent successes of neural models in other domains (He et al., 2015; Devlin et al., 2019), the publicly available LTR datasets are comparatively small. Popular techniques such as data augmentation to mitigate overfitting in high-capacity networks are commonly used in other areas (Perez & Wang, 2017). But it is less intuitive on how to do data augmentation for LTR datasets, compared with, e.g., rotating a cat image in computer vision.

### 4.2 IMPROVEMENTS

We introduce our proposed neural LTR framework that tries to address the above mentioned concerns. Figure 1 summarizes our DASALC framework, which stands for Data Augmented Self-Attentive Latent Cross ranking network.

#### 4.2.1 EXPLICIT FEATURE TRANSFORMATION AND DATA AUGMENTATION

Features in LTR datasets are diverse and can be of different scales. Out of the three datasets we consider, only the Yahoo dataset has been normalized (we leave it not-transformed). It is well known that neural networks are sensitive to input data scale, and we apply a simple "log1p" transformation to every element of $\mathbf{x}$ and empirically find it works well for the Web30K and Istella datasets:

$$\mathbf{x} = \log_e(1 + |\mathbf{x}|) \odot \text{sign}(\mathbf{x}). \tag{6}$$

where $\odot$ is the element-wise multiplication operator.

We use a very simple data augmentation technique on LTR datasets. We add a random Gaussian noise independently to every element of input vector $\mathbf{x}$:

$$\mathbf{x} = \mathbf{x} + \mathcal{N}(\mathbf{0}, \, \sigma^2 \mathbf{I}) \tag{7}$$

---

[3]We use the Fold1 in Web30K to be consistent with the setup of Yahoo and Istella. Some of the reported results on Web30K were based on 5-fold cross-validation (CV). We verified on $\lambda\text{MART}_{RankLib}$ that the difference between Fold1 and CV is small and does not affect our conclusion.

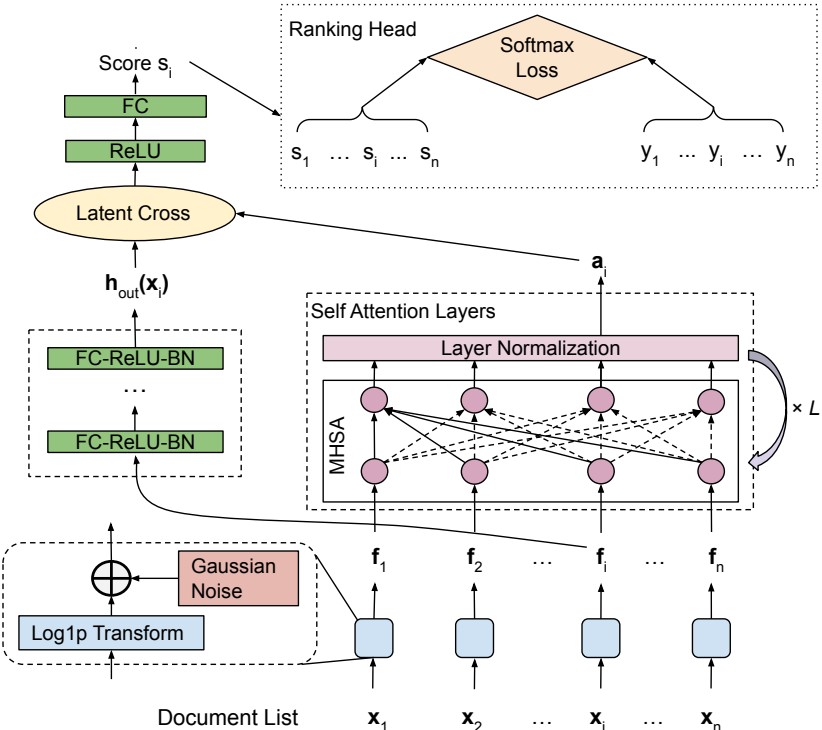

Figure 1: An illustration of the DASALC. FC is fully connected layer, ReLU is ReLU activation, and BN indicates batch normalization. Log1p Transform is applied when applicable. Softmax loss is short for softmax output with cross-entropy loss.

where $\sigma$ is a scalar hyperparameter. The random noise is added after the log1p transformation in an online fashion during training (i.e. different perturbations will be added to the same data point seen in different batches). A single scalar $\sigma$ for every feature is reasonable because the feature distributions are normalized by log1p. Also data augmentation is added after input Batch Normalization (BN) when applicable. Note that the random noise is added independently to every element so (later) BN will not cancel it away. We find such a simple data augmentation technique works well in our framework, but as shown in experiments, it only works when the capacity of the network is properly augmented as described in the next section.

For notation simplicity, we combine the log1p feature transformation and data augmentation into a single function $\mathbf{f} : \mathbb{R}^{n \times k} \to \mathbb{R}^{n \times k}$:

$$\mathbf{f} = \log_e(1 + |\mathbf{x}|) \odot \operatorname{sign}(\mathbf{x}) + \mathcal{N}(\mathbf{0}, \sigma^2 \mathbf{I}) \qquad (8)$$

### 4.2.2 LEVERAGING LISTWISE CONTEXT

For LTR problem, the list of documents can be leveraged in neural models. This is the key base to enhance the network architecture for LTR. We leverage the multi-head self-attention (MHSA) mechanism (Vaswani et al., 2017) to encode ranking list information. More specifically, we generate a contextual embedding $\mathbf{a}_i$, for each item $i$, considering the document similarity between document $i$ and every document in the list. For the multi-head self-attention mechanism, we have the input $\mathbf{f} \in \mathbb{R}^{n \times k}$, and project $\mathbf{f}$ into a query (in the context of attention mechanism) matrix $Q = \mathbf{f}W^Q$, a key matrix $K = \mathbf{f}W^K$, and a value matrix $V = \mathbf{f}W^V$ with trainable projection matrices $W^Q$, $W^K$, and $W^V \in \mathbb{R}^{k \times z}$, where $z$ is the attention head size. Then a self-attention (SA) head computes the weighted sum of the transformed values $V$ as,

$$\mathrm{SA}(\mathbf{f}) = \mathrm{Softmax}(S(\mathbf{f}))V, \qquad (9)$$

where similarity matrix between $Q$ and $K$ is defined as $S(\mathbf{f}) = \frac{QK^T}{\sqrt{z}}$. For each layer, the results from the $H$ heads are concatenated to form the output of multi-head self-attention by

$$\text{MHSA}(\mathbf{f}) = \text{concat}_{h \in [H]}[\text{SA}_h(\mathbf{f})]W_{\text{out}} + b_{\text{out}}, \tag{10}$$

where $W_{\text{out}} \in \mathbb{R}^{Hz \times z}$ and $b_{\text{out}} \in \mathbb{R}^{n \times z}$ are trainable parameters. We apply $L \geq 1$ layers of multi-head self-attention followed by a layer normalization (Ba et al., 2016) similarly to (Vaswani et al., 2017).

By treating $\mathbf{a}_i$ as the listwise contextual embedding for item $i$, we further leverage the simple latent cross idea (Beutel et al., 2018) to effectively generate feature interactions:

$$h_i^{\text{cross}} = (1 + \mathbf{a}_i) \odot h_{\text{out}}(x_i), \tag{11}$$

where $\odot$ is the element-wise multiplication operator ($\mathbf{a}_i$ will go through a linear projection when the dimensions do not match, omitted in the equation), and $h_{\text{out}}(x_i)$ is the output of the final hidden layer of regular network.

Learning to rank can be seen as learning to induce order over **set** of items. One desirable property for ranking approaches that use listwise context is to be *permutation equivariant*: applying a permutation over input items leads to an equivalent permutation over output scores. DASALC satisfies such a permutation equivariance property.

**Proposition 1.** *Let $\pi$ be a permutation of indices of $[1, .., n]$ and $\boldsymbol{x} \in \mathbb{R}^{n \times k}$ be the input item representation. DASALC is permutation equivariant for scores generated over input items , i.e, $s_{DASALC}(\pi(\boldsymbol{x})) = \pi(s_{DASALC}(\boldsymbol{x}))$. See proof at Appendix C.*

### 4.3 REMARKS

We compared several popular pointwise, pairwise, and listwise ranking losses. We report all results based on the softmax cross entropy loss $l(\mathbf{y}, s(\mathbf{x})) = -\sum_{i=1}^{n} y_i \log_e \frac{e^{s_i}}{\sum_j e^{s_j}}$ since it is simple and empirically robust in general, as demonstrated in Appendix B.2.

We provided a general framework that can enhance neural LTR models in many components. For each component, we purposefully use simple or well-known techniques for enhancement because the scope of the current research is to identify the possible reasons why neural LTR is under-performing when compared with the best traditional tree-based methods. Clearly, each component can use more advanced techniques, such as learning a more flexible data transformation (Zhuang et al., 2020) or using data augmentation policy (Cubuk et al., 2019), which we leave as future work.

## 5 EXPERIMENTS

We conduct experiments on the three LTR datasets (introduced in Sec 3.1) with our proposed framework and compare with some methods in Sec 3. For all our experiments using neural network approaches, we implemented them using the TF-Ranking (Pasumarthi et al., 2019) library.

We use two variants of our proposed approaches. DASALC is a model trained in our proposed framework. DASALC-ens is an ensemble of DASALC. By realizing LambdaMART is an ensemble method based on boosting, we leverage the randomness of neural model training and simply use the average score of 3-5 models (tuned on validation set) from different runs as the final score in DASALC-ens.

**Main result.** The results are summarized in Table 3. We focus on the comparison with $\lambda\text{MART}_{GBM}$ and also include SetRank to highlight the difference with recent neural LTR models. Readers can refer to Table 2 for more results. We tune hyperparameters on the validation sets, with more details in Appendix A. We have the following observations and discussions: (1) DASALC can sometimes achieve comparable or better results than $\lambda\text{MART}_{GBM}$, and outperforms recent neural LTR methods by a large margin. (2) DASALC-ens, though simple, can achieve neutral or significantly better results than $\lambda\text{MART}_{GBM}$ on all datasets and metrics. (3) The results on Yahoo dataset are weaker than the other two datasets. One thing to note is Yahoo dataset is already normalized upon release. As we note the importance of input feature transformation, the provided normalization may not be ideal for neural models, thus it should be encouraged to release LTR datasets with raw feature values.

Table 3: Result on the Web30K, Yahoo, and Istella datasets. $\uparrow$ means significantly better result, performanced against $\lambda\text{MART}_{GBM}$ at the $p < 0.05$ level using a two-tailed $t$-test. Last row is relative difference of DASALC-ens over $\lambda\text{MART}_{GBM}$.

| Models | Web30K NDCG@k | | | Yahoo NDCG@k | | | Istella NDCG@k | | |
|---|---|---|---|---|---|---|---|---|---|
| | @1 | @5 | @10 | @1 | @5 | @10 | @1 | @5 | @10 |
| $\lambda\text{MART}_{GBM}$ | 50.73 | 49.66 | 51.48 | **71.88** | **74.21** | **78.02** | **74.92** | 71.24 | 76.07 |
| SetRank$^{re}$ | 45.91 | 45.15 | 46.96 | 68.22 | 70.29 | 74.53 | 67.60 | 63.45 | 68.34 |
| DASALC | 50.95 | 50.92$^{\uparrow}$ | 52.88$^{\uparrow}$ | 70.98 | 73.76 | 77.66 | 72.77 | 70.06 | 75.30 |
| DASALC-ens | **51.89**$^{\uparrow}$ | **51.72**$^{\uparrow}$ | **53.73**$^{\uparrow}$ | 71.24 | 74.07 | 77.97 | 74.40 | **71.32** | **76.44**$^{\uparrow}$ |
| (Relative diff) | (+2.29%) | (+4.15%) | (+4.37%) | (-0.89%) | (-0.18%) | (-0.06%) | (-0.69%) | (+0.11%) | (+0.49%) |

Table 4: NDCG@5 on Istella when different components are added.

| Model | DNN | +log1p | +SA | +LC | +DA | +ens |
|---|---|---|---|---|---|---|
| NDCG@5 | 64.72 | 67.09 | 68.32 | 68.80 | 70.06 | 71.32 |

**Ablation study.** We provide some ablation study results in Table 4 to highlight the effectiveness of each component in our framework. Each component is added cumulatively from left to right in the table. We can see that each component helps and the best performance is achieved when all components are combined. More detailed ablation study is provided in Appendix B. Appendix B.1 gives more results on the effect of the log1p transformation. Appendix B.2 compares different loss functions and shows that listwise ranking loss performs better. Appendix B.3 shows the benefit of effective listwise context modeling. Appendix B.4 shows the effect of data augmentation in different model architectures.

# 6 RELATED WORK

We focus on traditional LTR problems when there are only numeric features and human ratings available. Some works (Mitra & Craswell, 2018; Nogueira et al., 2019; Han et al., 2020) on document matching and ranking leverage neural components such as word2vec and BERT when raw text is available, where the major benefit comes from semantic modeling of highly sparse input and tree-based methods become less relevant due to its limitation in handling sparse features.

The pioneering neural LTR models are RankNet (Burges et al., 2005) and LambdaRank (Burges et al., 2007). They use feed-forward networks on dense features as their scoring functions and became less favored than tree-based LambdaMART (Burges, 2010). Recent neural LTR models have explored new model architectures (Pang et al., 2020; Qin et al., 2020b), differetiable losses (Bruch et al., 2019b), and leveraging more auxiliary information (Ai et al., 2018). However, there is less work that specifically understands and addresses weaknesses for neural LTR, and a benchmark with strong tree-based baseline is missing. In this work, we show that relatively simple components that aim to address weaknesses of neural models can outperform recent methods significantly.

The idea of generating new data for LTR has been explored in few work recently, but their focus is to train more discriminative ranking models, not to mitigate the data sparsity problem for high-capacity neural models. For example, Yu & Lam (2019) uses a separate Autoencoder model to generate data and then feed them into tree-based models. This work can be treated as orthogonal to our data augmentation technique.

Several LTR papers have leveraged neural sequence modeling based on LSTM (Ai et al., 2018) or self-attention (Pang et al., 2020; Pasumarthi et al., 2020), which is not easy for tree-based approaches to model. We also leverage listwise context via self-attention to show neural LTR models are easily extendable. The combination of self-attention based listwise context and latent cross in our work to specifically mitigate the ineffectiveness of neural model to generate higher-order feature interactions has not been explored in the literature.

Our work is mostly orthogonal to another line of LTR research, namely unbiased learning to rank from implicit feedback data, such as clicks (Joachims et al., 2017; Hu et al., 2019; Qin et al., 2020a; Zhuang et al., 2021). There are also papers that try to reproduce tree models using neural architectures for tabular data (Saberian et al., 2019; Lee & Jaakkola, 2020). Our motivation is different in that our goal is to identify and mitigate weaknesses of neural approaches in general.

## 7 CONCLUSION AND DISCUSSION

In this paper, we first showed the inconsistency of performance comparison between neural rankers and GBDT models, and verified the inferior performance of neural models. We then identified the weaknesses when building neural rankers in multiple components and proposed methods to address them. Our proposed framework performs competitively well with the strong tree-based baselines. We believe our general framework and the rigorous benchmarking provides critical contribution to facilitate future neural LTR research. In particular, neural models are powerful in modeling complex relations (e.g, attention mechanism (Vaswani et al., 2017)) and raw text features (e.g., BERT (Devlin et al., 2019)). Also, the active research on neural networks in other domains continuously advances neural techniques (e.g., optimizers (Kingma & Ba, 2014)) All these can be studied in the LTR setting and our work pave ways to avoid pitfalls when leveraging these techniques.

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

# APPENDIX

## A  HYPERPARAMETER TUNING

For $\lambda \mathrm{MART}_{GBM}$, we do a grid search for number of trees $\in \{300, 500, 1000\}$, number of leaves $\in \{200, 500, 1000\}$, and learning rate $\in \{0.01, 0.05, 0.1, 0.5\}$.

For our neural models the main hyperparameters are hidden layer size $\in \{256, 512, 1024, 2048, 3072, 4096\}$ and number of layers $\in \{3, 4, 5, 6\}$ for regular DNN, data augmentation noise $\in [0, 5.0]$ using binary search with step 0.1, number of attention layers $\in \{3, 4, 5, 6\}$, and number of attention heads $\in \{2, 3, 4, 5\}$. The same parameter swept is enabled on the baselines we tried when applicable. One noticeable difference between our work and existing work is that we tried large hidden layer size up to 4096 and found that large models work better in general when data augmentation is enabled. We are in the process to release the code and trained models in an open-sourced software package.

## B  ABLATION STUDIES AND ANALYSIS

### B.1  EFFECT OF LOG1P INPUT TRANSFORMATION

We first show that the simple log1p transform can improve performance on the Web30K and Istella datasets (Yahoo dataset has already been normalized). Results in Table 5 are based on regular DNN models using the softmax cross-entropy loss. The trends are similar for other configurations. We also noted the results are in general slightly better than Gaussian normalization due to the long-tail nature of LTR dataset features, which we omit here.

| Method | Web30K NDCG@k | | | Istella NDCG@k | | |
|---|---|---|---|---|---|---|
| | @1 | @5 | @10 | @1 | @5 | @10 |
| Without log1p | 44.60 | 44.99 | 47.22 | 67.19 | 64.72 | 70.12 |
| With log1p | **48.30** | **48.22** | **50.35** | **69.78** | **67.09** | **72.49** |

Table 5: Results on Web30K and Istella using log1p input transformation.

We can see that such simple transformation can bring meaningful gains. In all following sections, we use log1p transformation by default.

## B.2 RANKING LOSSES

Many recent progresses of neural LTR are on ranking losses, especially listwise ranking losses (Bruch et al., 2019b;a; 2020; Grover et al., 2019). For example, it is attractive to devise differentiable versions of ranking losses for end-to-end learning. Here we do a benchmark of different ranking losses on regular DNN models on different datasets to show that (1) Listwise ranking losses are superior choices to pointwise or pairwise losses that are normally used for non-neural LTR models; (2) Performances of state-of-the-art listwise ranking losses are comparable; (3) The softmax cross entropy loss is a simple but robust choice.

We consider the following ranking losses:

- SigmoidCrossEntropy: a widely used pointwise loss: $l(\mathbf{y}, s(\mathbf{x})) = \sum_{i=1}^{n} -y_i s_i + \log_e(1 + e^{s_i})$.

- RankNet (Burges et al., 2005): a popular pairwise loss: $l(\mathbf{y}, s(\mathbf{x})) = \sum_{y_i > y_j} \log_e(1 + e^{s_j - s_i})$.

- LambdaRank (Burges et al., 2007; Wang et al., 2018b): the pairwise loss with $\Delta$NDCG weight, which is a direct implementation of the LambdaMART loss in Eq. (5).

- Softmax (Cao et al., 2007; Bruch et al., 2019a): a popular listwise loss: $l(\mathbf{y}, s(\mathbf{x})) = -\sum_{i=1}^{n} y_i \log_e \frac{e^{s_i}}{\sum_j e^{s_j}}$.

- ApproxNDCG (Qin et al., 2010; Bruch et al., 2019b): a listwise loss that is a differentiable approximation of NDCG metric: $l(\mathbf{y}, s(\mathbf{x})) = -\frac{1}{DCG(\pi^*, \mathbf{y})} \sum_{i=1}^{n} \frac{2^{y_i}-1}{\log_2(1+\pi_s(i))}$, where $\pi_s(i) = \frac{1}{2} + \sum_j \text{sigmoid}(\frac{s_j - s_i}{T})$ with $T$ a smooth parameter.

- GumbelApproxNDCG (Bruch et al., 2019b; 2020): a listwise loss with a stochastic treatment on ApproxNDCG: scores $s$ in the above NDCG loss function will be substituted by $s_i + g_i$, with a gumbel noise $g_i = -\log_e(-\log_e U_i)$ from $U_i$ uniformly sampled in $[0, 1]$.

- NeuralSortNDCG(Grover et al., 2019): a listwise loss that approximates NDCG metric with the NeuralSort trick: $l(\mathbf{y}, s(\mathbf{x})) = -\frac{1}{DCG(\pi^*, \mathbf{y})} \sum_{i,r=1}^{n} \frac{(2^{y_i}-1)P_{ir}^s}{\log_2(1+r)}$, where $P_{ir}^s$ is an approximate permutation matrix, obtained by NeuralSort trick: $P_{ir}^s = \text{softmax}[((n+1-2i)s_r - \sum_j |s_r - s_j|)/T]$, with $T$ a smooth parameter.

- GumbelNeuralSortNDCG: a listwise loss with a stochastic treatment of NeuralSortNDCG by replacing the score $s$ in neural sort permutation matrix by $s_i + g_i$, where $g_i$ is again sampled from the gumbel distribution. This is new in the literature but not the major focus of this work.

| Ranking loss | Web30K NDCG@k | | | Istella NDCG@k | | |
|---|---|---|---|---|---|---|
| | @1 | @5 | @10 | @1 | @5 | @10 |
| SigmoidCrossEntropy | 47.65 | 46.85 | 48.47 | 67.62 | 64.46 | 69.51 |
| RankNet | 46.05 | 46.67 | 48.98 | 69.09 | 66.04 | 71.81 |
| LambdaRank | 45.87 | 46.55 | 48.85 | 68.18 | 65.22 | 70.88 |
| Softmax | 48.30 | 48.22 | **50.35** | 69.78 | 67.09 | **72.49** |
| ApproxNDCG | 49.31 | 47.87 | 49.49 | 70.05 | 66.08 | 70.76 |
| GumbelApproxNDCG | 49.53 | 48.07 | 49.75 | **71.78** | **67.33** | 71.79 |
| NeuralSortNDCG | 48.66 | 47.19 | 48.83 | 68.92 | 64.27 | 69.03 |
| GumbelNeuralSortNDCG | **49.74** | **48.40** | 50.22 | 70.96 | 67.26 | 71.92 |

Table 6: Results on the Web30k and Istella datasets with standard feed-forward network architecture.

The results are summarized in Table 6. For different ranking losses, we make a grid search over different optimizers with different learning rates: for Adam optimizer, we scan learning rates $\in \{10^{-4}, 10^{-3}, 10^{-2}\}$; for Adagrad optimizer, we scan learning rates $\in \{0.01, 0.1, 0.5\}$. When the smooth parameter $T$ is applicable, we also scan it $\in \{0.1, 1, 10\}$. We report the results based on best NDCG@5 for different losses.

As we have stated above, we find that: (1) The performance of models trained with listwise losses are significantly better than the models trained with pointwise or pairwise losses. (2) Different

listwise losses are generally comparable, and we found that the softmax cross-entropy loss performs coherently well over different models and different datasets. It is thus used in our main results and following sections. (3) LambdaRank does not work well for neural models. On the other hand, previous work (Bruch et al., 2019a) shows that tree-based models with softmax loss are not as good as LambdaMART, demonstrating that tree-based models and neural LTR models have different behavior on different loss functions. This encourages future work to design neural LTR specific ranking losses.

## B.3 Effect of listwise context

We study the effect of leveraging listwise context with self-attention, with and without latent cross (concatenation between item feature and context feature will be applied) (Pasumarthi et al., 2020) on the Web30K and Istella datasets. Results are shown in Table 7. We can see that using neural approach to model listwise context, which is difficult for tree-based models to do, is quite beneficial. Latent cross, though simple, can help leverage listwise context more effectively.

| Method | Web30K NDCG@k | | | Istella NDCG@k | | |
|---|---|---|---|---|---|---|
| | @1 | @5 | @10 | @1 | @5 | @10 |
| DNN | 48.30 | 48.22 | 50.35 | 69.78 | 67.09 | 72.49 |
| Self-attention | 49.89 | 50.15 | 52.18 | 71.77 | 68.32 | 73.72 |
| Self-attention and LC | **50.19** | **50.49** | **52.47** | **72.19** | **68.80** | **74.21** |

Table 7: Results on the Web30K and Istella datasets using self-attention and latent cross.

## B.4 Effect of data augmentation

One of the technical findings in this work is that using a simple Gaussian noise as data augmentation can help neural LTR models. Below we add Gaussian noise with different strength ($\sigma$) to both DNN model and the DASALC framework with results shown in Table 8. We can see that the performance

| Method ($\sigma$) | Web30K NDCG@k | | |
|---|---|---|---|
| | @1 | @5 | @10 |
| DNN(0.0) | 48.30 | 48.22 | 50.35 |
| DNN(0.1) | 48.10 | 48.01 | 50.14 |
| DNN(1.0) | 46.39 | 46.15 | 48.18 |
| DASALC(0.0) | 50.19 | 50.49 | 52.47 |
| DASALC(0.1) | 50.38 | 50.61 | 52.56 |
| DASALC(1.5) | **50.95** | **50.92** | **52.88** |
| DASALC(2.0) | 50.65 | 50.78 | 52.68 |

Table 8: Results on the Web30K datasets using different architecture and random noise strength.

of DNN starts to drop as soon as we start to add noise. However, for DASALC, data augmentation helps and the performance looks robust using different levels of noise. The performance peeks around $\sigma = 1.5$. The optimal $\sigma$ needs tuning for different datasets but the general trends are similar for other datasets. We treat the study of the exact mechanism of how data augmentation works in DASALC and the application of more sophisticated data augmentation techniques as future work.

We also try to add noise to $\lambda$MART$_{GBM}$ and see similar results as DNN. The results on the YAHOO dataset is shown in Table 9, we can see that adding noise leads to worse accuracy.

| Method ($\sigma$) | Yahoo NDCG@k | | |
|---|---|---|---|
| | @1 | @5 | @10 |
| $\lambda$MART$_{GBM}$(0.0) | 71.88 | 74.21 | 78.02 |
| $\lambda$MART$_{GBM}$(0.1) | 70.10 | 72.60 | 77.19 |
| $\lambda$MART$_{GBM}$(1.0) | 64.96 | 67.28 | 72.60 |
| $\lambda$MART$_{GBM}$(1.5) | 64.39 | 66.84 | 72.27 |

Table 9: Results on the Yahoo datasets using different architecture and random noise strength.

### B.5 Performance on Catboost

We mainly compared with $\lambda\text{MART}_{RankLib}$ and $\lambda\text{MART}_{GBM}$ in the main content since they are the most popular baselines used in recent papers. There are other GBDT implementations that can also be used for the LTR task. Catboost (Prokhorenkova et al., 2018) is a recently popular GBDT implementation for various tasks. We also evaluate its performance on the three LTR datasets. Note that Catboost is not specific to ranking and does not have a standard LambdaMART implementation to the best of our knowledge. We try both the QueryRMSE loss and YetiRank loss, which are the best performing losses on most existing Catboost's benchmarks. The results are reported in Table 10.

Table 10: Comparison of Catboost with other methods on the Web30K, Yahoo, and Istella datasets.

| Models | Web30K NDCG@k | | | Yahoo NDCG@k | | | Istella NDCG@k | | |
|---|---|---|---|---|---|---|---|---|---|
| | @1 | @5 | @10 | @1 | @5 | @10 | @1 | @5 | @10 |
| $\lambda\text{MART}_{RankLib}$ | 45.35 | 44.59 | 46.46 | 68.52 | 70.27 | 74.58 | 67.71 | 61.18 | 65.91 |
| $\lambda\text{MART}_{GBM}$ | 50.73 | 49.66 | 51.48 | 71.88 | 74.21 | 78.02 | 74.92 | 71.24 | 76.07 |
| Catboost-QueryRMSE | 50.07 | 50.04 | 51.97 | 70.50 | 74.25 | 78.31 | 69.91 | 67.73 | 72.18 |
| Catboost-YetiRank | 48.92 | 49.10 | 51.31 | 69.86 | 74.00 | 78.11 | 72.06 | 69.97 | 74.12 |
| DASALC-ens | 51.89 | 51.72 | 53.73 | 71.24 | 74.07 | 77.97 | 74.40 | 71.32 | 76.44 |

We can see that Catboost can produce very decent results, clearly outperforming $\lambda\text{MART}_{RankLib}$, but its comparison with $\lambda\text{MART}_{GBM}$ is mixed. We encourage researchers to also consider different implementations such as Catboost in future LTR work.

### B.6 LambdaMART Ensemble

We showed that a simple ensemble of neural rankers can bring meaningful gains, leveraging the stochastic nature of neural network learning. On the other hand, LambdaMART itself is an ensemble algorithm using boosting, but it is still interesting to see the effect of ensembling multiple LambdaMART models. We conduct additional experiments on this front using $\lambda\text{MART}_{GBM}$ and have two major observations: 1) Running LambdaMART multiple times with the same configuration generates very similar results, and ensemble in this setting does not help, whereas neural rankers can benefit from such a simple setting; 2) In Table 11 we show ensembling LambdaMART with different configurations (e.g., different # trees, # leaves and learning rate) on the Istella dataset. We ensemble five LambdaMART models chosen on the validation set. The results on other datasets are similar.

| Method | Istella NDCG@k | | |
|---|---|---|---|
| | @1 | @5 | @10 |
| $\lambda\text{MART}_{GBM}$ | 74.92 | 71.24 | 76.07 |
| $\lambda\text{MART}_{GBM}$-ens | 75.04 | 71.40 | 76.28 |

Table 11: Results on the Istella datasets using LambdaMART ensembles.

We can see that the improvement from ensembling LambdaMART is smaller than that in neural rankers (see Table 3). Our hypothesis is that model ensembles tend to be more effective for neural rankers with stronger stochastic nature, and exploring advanced model ensemble methods with neural rankers is an interesting future direction.

## C Permutation equivariance analysis

For any general scoring function $s(\mathbf{x}) : \mathbb{R}^{n \times k} \to \mathbb{R}^n$, and a permutation $\pi$ over indices $[1, ..., n]$, we call $s$ to be permutation equivariant iff

$$s(\pi(\mathbf{x})) = \pi(s(\mathbf{x})) \tag{12}$$

The scoring function for proposed approach, DASALC, can be written as a combination of feature transformation and data augmentation function $\mathbf{f}$, output of multi-headed self-attention $\mathbf{a} :=$

MHSA$^L(\mathbf{f})$ and output of final layer of regular network $h_{out}(\mathbf{x})$.

$$s_{DASALC}(\mathbf{x}) = W_{FC}^T ReLU((1 + \mathbf{a}(\mathbf{x})) \odot h_{out}(\mathbf{x})) \tag{13}$$

Note that per-item transformations, which we refer to as *univariate* transformations, are trivially permutation equivariant. Also, composition of two permutation equivariant functions is also permutation equivariant, as the permutation operator and the permutation equivariant functions are commutative. Hence linear projection, ReLU activation and $\mathbf{f}$ (as a function of $\mathbf{x}$) are permutation equivariant. Multi-headed self-attention is shown to be permutation equivariant (Pang et al., 2020). Hence, on applying permutation $\pi$ to the proposed scoring function, we see that it satisfies the permutation equivariance property.

$$
\begin{aligned}
\pi(s_{DASALC}(\mathbf{x})) &= \pi(W_{FC}^T \, ReLU((1 + \mathbf{a}(\mathbf{x})) \odot h_{out}(\mathbf{x}))) \\
&= W_{FC}^T \, ReLU(\pi((1 + \mathbf{a}(\mathbf{x})) \odot h_{out}(\mathbf{x}))) \\
&= W_{FC}^T \, ReLU((1 + \pi(\mathbf{a}(\mathbf{x})) \odot \pi(h_{out}(\mathbf{x}))) \\
&= W_{FC}^T ReLU(((1 + \mathbf{a}(\pi(\mathbf{x}))) \odot h_{out}(\pi(\mathbf{x})))) \\
&= s_{DASALC}(\pi(\mathbf{x}))
\end{aligned}
$$

