# OpenReview forum: "Are Neural Rankers still Outperformed by Gradient Boosted Decision Trees?"
_ICLR.cc/2021/Conference — ICLR 2021 Spotlight_

### Official Review · AnonReviewer1 · 2020-10-20
**Much needed work for neural LTR research to make progress**

**Rating:** 8
**Confidence:** 4

**Review:**

Originality, Significance: This paper establishes reference points for modern LTR research. The fact that RankLib is a very popular but also a weak baseline has been exploited by too many researchers for too long. When I was reviewing other LTR papers, I often had to point out that the proposed method significantly underperforms LightGBM. This fact has been fairly well known, but apparently not widespread enough. Having an ICLR paper published on this issue will help spreading the fact, which is significant on its own.

Quality: Considering the popularity of RankLib, deeper analysis of why LightGBM outperforms RankLib would've been very nice, however. Authors do mention that LightGBM has more features, but it is unclear which exact feature of LightGBM contributed to such a significant difference between the two. Understanding the reason for LightGBM's superiority could potentially help us to develop better LTR models, neural or not neural. Comparison against Catboost https://github.com/catboost/catboost would've also been useful, as it is often claimed to outperform LightGBM.

Proposed DASALC framework is quite simple and uses mostly standard techniques, and this is an advantage as a reference point. Still, DASALC significantly outperforms previous neural LTR approaches. Also, although the idea of applying these standard techniques on LTR seems straightforward, but I argue that's only due to the benefit of hindsight; neural LTR has been a fairly active area of research, yet these techniques haven't yet been widely used in LTR literature. It would've been very interesting to see how these techniques improve the performance of previous neural LTR models; log1p transformation, data augmentation, and model ensembling would straightforwardly apply to other neural models as well.

In summary, I believe this paper will foster more productive research by establishing the strong baseline on both decision-tree based method (although it has been known) and neural method (on which authors make good technical contributions).

Clarity: The paper is quite easy to follow.

---

> ### Author Response · Authors · 2020-11-22
> **Reply to AnonReviewer1**
>
> Thank you very much for your kind words and feedback. We appreciate your recognition of  the importance of this work and the current situation in the LTR community.
>
> We agree that understanding why LightGBM works is an important direction to help LTR in general, and it can be a great topic on its own. We only included RankLib and LightGBM since they are most popular and there is a huge gap between them that led to the inconsistency in the literature. We were aware of Catboost, note that Catboost is a GBDT implementation but not specific to ranking, and it does not have the LambdaMART implementation if we understand correctly. We tried the QueryRMSE loss and YetiRank loss, which perform best on most existing Catboost’s benchmarks. On our concerned datasets below, we can see Catboost seems to show mixed results when compared with LightGBM but it still outperforms RankLib consistently. We believe this makes the main information in the paper intact. We added this result and discussion to the appendix.
>
> Web30k:
>
> |                     | NDCG@1 | NDCG@5 | NDCG@10 |
> |---------------------|--------|--------|---------|
> | RankLib             | 45.35  | 44.59  | 46.46   |
> | LightGBM            | 50.73  | 49.66  | 51.48   |
> | CatBoost- QueryRMSE | 50.07  | 50.04  | 51.97   |
> | Catboost- YetiRank  | 48.92  | 49.10  | 51.31   |
> | DASALC-en           | 51.89  | 51.72  | 53.73   |
>
>
> ---
> Yahoo:
>
> |                     | NDCG@1 | NDCG@5 | NDCG@10 |
> |---------------------|--------|--------|---------|
> | RankLib             | 68.52  | 70.27  | 74.58   |
> | LightGBM            | 71.88  | 74.21  | 78.02   |
> | CatBoost- QueryRMSE | 70.50  | 74.25  | 78.31   |
> | Catboost- YetiRank  | 69.86  | 74.00  | 78.11   |
> | DASALC-en           | 71.24  | 74.07  | 77.97   |
>
> ---
> Istella:
>
> |                     | NDCG@1 | NDCG@5 | NDCG@10 |
> |---------------------|--------|--------|---------|
> | RankLib             | 65.71  | 61.18  | 65.91   |
> | LightGBM            | 74.92  | 71.24  | 76.07   |
> | CatBoost- QueryRMSE | 69.91  | 67.73  | 72.18   |
> | Catboost- YetiRank  | 72.06  | 69.97  | 74.12   |
> | DASALC-en           | 74.40  | 71.32  | 76.44   |
>
> ---
> We agree that the proposed techniques are general and that’s the major goal. We actually tried the techniques on both standard NN and models with self-attention mechanism, which are backbones of many recent neural papers, showing their generality. We will see what other models we can consider and encourage the community to try and develop more advanced techniques with respect to each module.

---

### Official Review · AnonReviewer2 · 2020-10-23
**Great focused contribution for ranking**

**Rating:** 8
**Confidence:** 4

**Review:**

Thanks to the authors for their hard work and the nice paper. I both enjoyed this paper and think it's a strong contribution to the ranking literature. It was well written, clear, and nicely organized. The appendices are full of useful experiments.

There is some room for improvement, particularly in the thoroughness of the experiments:

- Eq (6) - It's nice to have a single transform for everything but what if the data is already, say, standard normal? Or categorical? Do any of your benchmark datasets have categorical variables? How were they dealt with?
- Sec 3.2 - Hyperparameters for neural networks - learning rate and batch size are usually crucial for neural networks, but are not tuned for any of of the baselines (as far as I could tell). You are likely to get higher performance if these are tuned as well - please do so, especially for the baselines that you implemented yourself.
- It appears from the ablation study that some of the the bigger performance boosts came from the feature transformation and data augmentation. As this was meant to address issues with the benchmark methods, it would be very worthwhile to apply these techniques to those methods, perhaps while increasing the network capacity.
- Sec 5 Main Result - for fairness, you should compare to an ensemble of lambdaMARTs.
- What about training time? LightGBM is very fast. How long does it take to train your model? Is the few percent improvement worth it?
- Does lambdaMART also get better with Gaussian noise data augmentation?

One final note: I'm not sure the word "hitherto" makes sense in the title.

---

> ### Author Response · Authors · 2020-11-22
> **Reply to AnonReviewer2**
>
> Thank you very much for your kind words and feedback.
>
> This paper focuses on the long-standing traditional LTR problem with public datasets where only numeric features are available (a setting that has been used in numerous papers), which we limit the scope of this work to. Our experience is it may be easier for neural models to handle sparse/categorical data by, e.g., using embeddings without transformation, but it is out of the scope of this paper.
>
> ---
> We did extensive tuning of hyperparameters of the baselines we ran by ourselves. We added  some clarification of the hyperparameter tuning of baselines in Appendix A.
>
> ---
> We agree that the proposed modules can be applied to neural rankers in general. In fact in the paper we show that these techniques work for standard feedforward NN and NN with attention mechanism, which are backbones of numerous recent neural rankers, e.g., with different loss functions, a focus of many LTR papers. We will try to add more results in a future version.
>
> ---
> Thank you for your feedback. We mentioned in the paper that lambdaMART itself is an ensemble. Our neural ranker ensemble takes the advantage of stochastic nature of neural learning so different checkpoints in the same or different runs with the same configuration can be easily combined together using ensemble (we simply use the mean score of different models, despite more advanced techniques such as boosting). In terms of model ensembles for LambdaMART, we ran some additional experiments and got two observations: 1) Running lambdaMART multiple times with the same configuration will generate very similar results, so ensemble in this setting does not help, whereas neural rankers can benefit from such a simple setting. 2) Below we show that assembling lambdaMART with different configurations (e.g., different # trees, # leaves and learning rate) on the Istella dataset. The results on other datasets are similar.
>
>
> |              | NDCG@1 | NDCG@5 | NDCG@10 |
> |--------------|--------|--------|---------|
> | LightGBM     | 74.92  | 71.24  | 76.07   |
> | LightGBM-ens | 75.04  | 71.40  | 76.28   |
>
>
> We can see that the improvement from ensemble is much smaller than that in neural rankers, where we get >1.0 gain. Model ensembles tend to be more effective for neural rankers with better stochastic nature. We added the new results and discussion to the appendix.
>
> ---
> It takes several hours to train the neural models on modern hardware so it is manageable. The public LightGBM implementation takes a couple of days to finish a run on a single machine on the Istella dataset. We are aware that LightGBM provides parallel computation support but did not run it. In terms of if extra running time (if there are any) is worth it, we believe this paper will encourage more advanced neural LTR research that will boost the performance even further in the future, under similar manageable time budgets.
>
> We tried to add noise to lambdaMART (in LightGBM) and it actually hurts performance, see the table below for  the results on the Yahoo dataset. Results on other datasets are similar. This is reasonable since there does not seem to be much evidence in the literature showing data augmentation helping tree-based models.
>
>
> |                              | NDCG@1 | NDCG@5 | NDCG@10 |
> |------------------------------|--------|--------|---------|
> | LightGBM                     | 71.88  | 74.21  | 78.02   |
> | LightGBM with variance = 0.1 | 70.10  | 72.60  | 77.19   |
> | LightGBM with variance = 1.0 | 64.96  | 67.28  | 72.60   |
> | LightGBM with variance = 1.5 | 64.39  | 66.84  | 72.27   |
>
>
> ---
> In terms of the title, we can remove “hitherto” or replace it with “still”: “Neural Rankers are still Outperformed by Gradient Boosted Decision Trees?”. Any suggestions will be appreciated.

---

### Official Review · AnonReviewer4 · 2020-10-27
**Unfair comparison and limited novelty**

**Rating:** 2
**Confidence:** 5

**Review:**

- The paper argues that neural models perform significantly worse than GBDT models on some learning to rank benchmarks. It first conducts a set of experiments to show that GBDT outperforms some neural rankers. Then, it presents a few tweaks related to feature transformation and data augmentation to improve the performance of neural models. The resulting neural models perform on par with the state-of-the-art GBDT models.
- I think this paper establishes an unfair comparison between GBDT and neural-based models. As known,  neural models are good at learning great representations from the raw inputs, such as audio, images, and texts, while GBDT models are good at dealing with sparse features. A more fair comparison could be having the neural models to learn feature representations, which will be concatenated with the normalized sparse features.
- Finally, the technical contribution of the paper is also quite limited.

---

> ### Author Response · Authors · 2020-11-22
> **Reply to AnonReviewer4**
>
> Thank you very much for your feedback.
>
> Re: “an unfair comparison between GBDT and neural-based models. As known, neural models are good at learning great representations from the raw inputs, such as audio, images, and texts”.
>
> We agree on the strength of neural models in learning from raw inputs like texts. Indeed, BERT-based neural models are already dominating text-only document ranking and researchers already reached consensus [1]. But in LTR settings, there are still many traditional features like TF-IDF and BM25, which are the accumulation of many years of research in Information Retrieval. Both raw input and traditional features are used in practical LTR models [2]. Improvement on either of them can be translated to the combined models. However, research work on how to better leverage numerical features, especially in neural models, is still lacking for LTR models.
>
> The review points out a reasonable approach to ensemble neural and tree models by stacking them when both raw inputs and traditional features are available. However, our current research can still be beneficial in this setting by feeding the traditional features to neural models to get better representations of items before stacking tree models on the top.
>
> As we pointed out in the paper, commonly used benchmarks for LTR mostly have traditional features and are still heavily used by the research community. We hope that our work can set a good base for future research on LTR benchmark datasets since any progress on this front can be translated to the ensemble models when both traditional features and raw inputs are available.
>
> ---
> Additional response from authors:
>
> We believe that there is some misunderstanding of the scope of this work. If the reviewer’s concern is on comparing NN and GBDT head-to-head with the same inputs, we argue that the exact setting studied in this paper is very popular with many papers published recently. But as pointed out in the paper, the fact is that neural rankers have been underperforming in this setting and the claimed progress in recent publications needs to be validated. This is causing severe concerns in the community as pointed out by AnonReviewer1. The goal of this paper is to rigorously show that neural rankers can actually work well on this important problem, bridging the big gap and correcting several misunderstandings in the literature.
>
>
> [1] Lin, Jimmy, Rodrigo Nogueira, and Andrew Yates. "Pretrained Transformers for Text Ranking: BERT and Beyond." arXiv preprint arXiv:2010.06467 (2020).
>
> [2] Yu Meng, Maryam Karimzadehgan, Honglei Zhuang, Donald Metzler. Separate and Attend in Personal Email Search. WSDM (2020).

---

### Official Review · AnonReviewer3 · 2020-10-29
**Review comment regarding #952**

**Rating:** 6
**Confidence:** 3

**Review:**

Summary
In this paper, the authors study the problem of neural LTR models. They discuss why neural LTR models are worse than gradient boosted decision tree-based LTR models, and introduce some directions to improve neural LTR models.

Pros:
This paper discusses potential reasons why neural LTR models are worse than gradient boosted decision tree-based LTR models, and uses empirical results to show the effectiveness of the proposed solutions.

Concerns:
Regarding the proposed solutions, the authors use data augmentation to improve neural LTR models. It seems that some feature engineering work can help improve performance. Comparing to traditional gradient boosted tree-based LTR models, is it really worth putting efforts into studying neural LTR models?

---

> ### Author Response · Authors · 2020-11-22
> **Reply to AnonReviewer3**
>
> Thank you very much for your feedback.
>
> Feature engineering was largely overlooked in LTR in general due to the dominance of tree models in the past. There are mainly numerical features in LTR benchmark datasets and feature engineering does not seem to help much on tree models (note that we are referring to feature transformation, not generating new features). For example, while tree models excel at partitioning the input feature space regardless of feature scales, neural networks are more sensitive to feature scales. However, this has not been acknowledged in recent neural network based LTR. One point in this paper is to show that simple transformations can help neural rankers greatly and are encouraged to be adopted in the future research in this field so that the progress can be measured solidly.
>
> ---
> Though neural LTR models need special treatments, we believe that studying them is very important because: 1) The potential of neural LTR models is not yet fully realized. This is partly demonstrated by the recent active research in this area with many papers published; 2) To make solid progress by the research community in the field, we think benchmarking is critical. This paper points out the inconsistent comparison issue that has been hindering progress in the area; 3) This paper shows that neural rankers can work well by addressing several bottlenecks that have been largely overlooked; 4) The paper proposes several general building blocks that could leverage more advanced techniques from the rich neural network literature, further pushing the limits of neural rankers; 5) In practice, if there was a neural model which can only achieve similar performance to a gradient boosted tree-based LTR model, it could still be valuable as it would simplify the overall infrastructure considering that there are often other neural models handling raw text/image features.

---

### Author Response · Authors · 2021-01-14
**Changing the title**

We are updating the title according to AnonReviewer2's comment.

---

### Decision · Program_Chairs · 2021-01-07
**Final Decision**

**Decision:**

Accept (Spotlight)

**Comment:**

Three reviewers are positive, while one is negative. The negative reviewer is well-qualified, but the review is not persuasive. Overall, this paper should be published as a wake-up call to the research community. Unfortunately, the lesson of this paper is similar to that of several previous papers, in particular

Armstrong, T. G., Moffat, A., Webber, W., & Zobel, J. (2009, November). Improvements that don't add up: ad-hoc retrieval results since 1998. In Proceedings of the 18th ACM conference on Information and knowledge management (pp. 601-610).

This submission should be a spotlight, to maximize the chance that future researchers learn its lesson.